# Choline Hydrogen Dicarboxylate Ionic Liquids by X-ray Scattering, Vibrational Spectroscopy and Molecular Dynamics: H-Fumarate and H-Maleate and Their Conformations

**DOI:** 10.3390/molecules25214990

**Published:** 2020-10-28

**Authors:** Simone Di Muzio, Fabio Ramondo, Lorenzo Gontrani, Francesco Ferella, Michele Nardone, Paola Benassi

**Affiliations:** 1Department of Physical and Chemical Sciences, University of L’Aquila, Via Vetoio, I-67100 L’Aquila, Italy; simone.dimuzio@libero.it (S.D.M.); francesco.ferella@lngs.infn.it (F.F.); michele.nardone@univaq.it (M.N.); paola.benassi@univaq.it (P.B.); 2Department of Chemistry, University of Rome La Sapienza, P.le A. Moro 5, 00185 Rome, Italy; lorenzo.gontrani@uniroma1.it; 3INFN, Gran Sasso National Laboratories, I-67100 Assergi (AQ), Italy

**Keywords:** ionic liquids, structure, molecular dynamics, X-ray diffraction, vibrational spectroscopy

## Abstract

We explore the structure of two ionic liquids based on the choline cation and the monoanion of the maleic acid. We consider two isomers of the anion (H-maleate, the *cis*-isomer and H-fumarate, the *trans*-isomer) having different physical chemical properties. H-maleate assumes a closed structure and forms a strong intramolecular hydrogen bond whereas H-fumarate has an open structure. X-ray diffraction, infrared and Raman spectroscopy and molecular dynamics have been used to provide a reliable picture of the interactions which characterize the structure of the fluids. All calculations indicate that the choline cation prefers to connect mainly to the carboxylate group through OH⋯O interactions in both the compounds and orient the charged head N(CH_3_)_3_^+^ toward the negative portion of the anion. However, the different structure of the two anions affects the distribution of the ionic components in the fluid. The trans conformation of H-fumarate allows further interactions between anions through COOH and CO_2_^−^ groups whereas intramolecular hydrogen bonding in H-maleate prevents this association. Our theoretical findings have been validated by comparing them with experimental X-ray data and infrared and Raman spectra.

## 1. Introduction

Protic Ionic Liquids (PILs) [1] are an important subset of the broad class of materials, called ionic liquids (ILs) [2,3,4,5,6,7], developed and studied in the last 20 years for a variety of applicative issues [8]. PILs are easily obtained by simple acid–base reactions; the resulting salt is usually liquid at room temperature or more generally below 100 °C. PILs with biodegradable and toxicological properties (BioILs) [9,10,11] have been developed, combining choline cations (Ch) with amino acid anions [12,13,14,15,16,17] or a large variety of carboxylic acids of natural origin [11,18,19]. The properties and applications of ionic liquids consisting of choline cations and carboxylate anions have been the subject of a wide number of studies [18] and their structures have been investigated by experiments and theories [20,21,22]. Despite the chemical simplicity of their synthesis, PILs show quite a complex structure where interactions between molecular constituents include hydrogen bonding along with long-range Coulombic and short-range van der Waals forces. The fundamental role of hydrogen bonding in determining the structure of BioILs has been shown in a series of systems where a choline cation is combined with amino acid anions [23,24,25,26]. Infrared and Raman spectra and X-ray studies on some choline-based BioILs coupled with carboxylate anions have been reported and their structures have been determined by classical and ab initio molecular dynamics [20,21,22]. Moreover, vibrational spectroscopy has been largely applied to investigate hydrogen bonding and its dynamic aspects in liquids, starting from pure water [27,28]. Among the choline-based BioILs, our group studied a carboxylate anion, salicylate, which is able to form strong intramolecular hydrogen bonding [22]. X-ray scattering, vibrational spectroscopy and molecular dynamics indicated that the carboxylate group is connected to choline by an OH group and simultaneously forms an intramolecular hydrogen bond with the OH group bonded to the benzene ring.

The aim of the present study is to elucidate the role of intramolecular hydrogen bonding in two BioILs constituted by the choline cation [Ch] coupled with the monoanionic maleic acid [H-maleate] and choline coupled with the monoanionic fumaric acid [H-fumarate]. H-maleate and H-fumarate are two isomers (cis and trans) of the HOOC − CH = CH − CO_2_^−^ anion. The cis-conformation (H-maleate) allows the formation of intramolecular hydrogen bonding whereas the trans-conformation (H-fumarate) hinders any intramolecular interaction. This structural difference affects the physical chemical properties of the respective choline ILs. The melting point of [Ch][H-maleate] (25 °C) is lower than [Ch][H-fumarate] (80 °C) [11], whereas, among the choline-based BioILs analyzed by Fukaya et al. [11], [Ch][H-maleate] was found to have the lowest viscosity value at room temperature. Both the properties have been interpreted as being due to the presence of intramolecular hydrogen bonding in H-maleate that weakens the electrostatic interactions with the cation. Consistently, the Kamlet-Taft parameters measured for some choline dicarboxylate monoanion ILs [11] revealed that the intramolecular hydrogen bond of [Ch][H-maleate] reduces the proton-accepting nature of the hydrogen maleate anion and weakens the hydrogen bond.

Intramolecular hydrogen bonding in H-maleate has also been observed in the gas phase [29] and in the crystal phase [30,31,32]. In particular inelastic neutron scattering [30], infrared spectroscopy and computational studies [31,32] indicated that the intramolecular hydrogen bond in H-maleate crystal is very strong and symmetric and, moreover, the intramolecular proton transfer potential is extremely shallow. In the present work, we want to compare the structure of the [Ch][H-maleate] and [Ch][H-fumarate] liquids through a combination of X-ray diffraction, infrared and Raman spectroscopy and molecular dynamics (MD) studies. The aim of this study is to evaluate if the different conformation of the H-fumarate and H-maleate anions can really affect the microscopic structure of the liquid.

## 2. Materials and Methods

### 2.1. Experimental Details

#### 2.1.1. Synthesis

Choline hydroxide solution (46 wt.% in H${}_{2}$O, Aldrich (Italy)), maleic acid (99%, Aldrich) and fumaric acid (99%, Aldrich) were the reagents used, without further purification, in the synthesis. Choline maleate was synthesized by dropwise addition of maleic acid to the choline hydroxide solution in a 1:1 ratio, stirring continuously at room temperature and pressure for 12 h. Most of the water in the reaction mixture was removed under reduced pressure, using a rotary evaporator at 70 °C for 4 h. The mixture was then dried in vacuo, with heating at 70 °C and stirring for 24 h. The same procedure was followed for the synthesis of choline fumarate by adding fumaric acid to the choline hydroxide solution. The purity of these BioILs was checked by ${}^{1}$H-NMR and ${}^{13}$C-NMR spectroscopy, using a Bruker Avance III spectrometer operating at 400 MHz and 100.6 MHz, respectively. The spectra have confirmed the absence of any major impurities and the water final content, evaluated by ${}^{1}$H-NMR analysis, has been estimated to be below 0.4 wt.%.

#### 2.1.2. X-ray Scattering

X-ray diffraction experiments were carried out with an energy-dispersive instrument that exploits the dependence of diffracted intensity on the energy of the radiation (EDXD—Energy-Dispersive X-ray Diffraction). More details on this method can be found in the supporting information and in previous papers [33,34,35,36,37], where the technique has been successfully employed in the study of non-crystalline systems (liquids, amorphous), and particularly to investigate the structure of ionic liquids [38]. In summary, the outcome of an EDXD experiment consists of a wide *q*-ranging structure factor and total radial distribution patterns that are related to the structural correlations existing among the particles of the system, and that can be modelled and interpreted using analogous patterns built from pairwise distribution functions (g(r)) obtained from simulations, like molecular dynamics.

#### 2.1.3. Infrared and Raman Spectra

Fourier transformed infrared (FTIR) spectra were measured at room temperature from 4000 to 400 cm${}^{-1}$ after 100 scans using the Perkin Elmer Spectrum two FT-IR spectrometer. Liquid samples were placed as thin films between KBr plates. Raman spectra were measured using a LABRAM confocal microscope Raman spectrometer by HORIBA Jobin Yvon using 5 mW at a 632-nm excitation source and a 20× collection optics. The instrumental resolution is of the order of 2–3 cm${}^{-1}$. Background fluorescence has been fitted using a polynomial expression and subtracted from the data.

### 2.2. Computational Details

The structure of the ionic liquids has been investigated following two theoretical approaches. Local structural ion pairs, with particular attention to the hydrogen bonding interaction, was characterized initially by ab initio methods. Quantum mechanical (QM) calculations on the [Ch][H-fumarate] and [Ch][H-maleate] ion pairs were performed using the Gaussian 09 package [39]. Equilibrium geometry and vibrational frequencies were obtained using density functional theory (DFT) methods with the B3LYP [40,41] exchange and correlation functional and employing the 6–311++G** basis set. The vibrational modes of the isolated ions and ion pairs have been assigned by the analysis of the Potential Energy Distribution (PED) using the VEDA software [42]. The good accuracy of such a functional in predicting the geometries of several organic molecules [43] and energetic properties of ionic liquid clusters [44] has been largely discussed in previous studies. Moreover, the stability and geometries of the most stable ion pairs were further studied at the MP2/6–311++G** level to evaluate the accuracy of the functional used for the DFT calculations in our systems.

Dynamic effects were introduced at ab initio level by Ab Initio Molecular Dynamics (AIMD) simulations performed on a model of 10 ion pairs using the Born–Oppenheimer Molecular Dynamics (BOMD) method implemented in the CP2K code [45]. Potential energy calculations were carried out using the BeckeLeeYangParr exchange correlation functional (BLYP) [41,46] and the hybrid Gaussian and plane wave (GPW) basis set. The Gaussian basis set was double zeta valence plus polarization functions optimized for molecular calculations at short range (DZVP-MOLOPT-SR) and the plane wave expansion was developed in a periodic cubic system with a unit cell edge of 15 Å${}^{3}$ and truncated at 320 Ry. Goedecker–Teter–Hutter (GTH) pseudopotentials [47,48] were employed to describe core electrons. A pre-equilibration was performed by employing classical molecular dynamics within periodic boundary conditions using the two-body Generalized Amber Force Field (GAFF)[49]. AIMD simulations started from an initial configuration of the classical MD simulation and the system was equilibrated for 6.3 ps in the NVT ensemble at 300 K using the individual thermostat for each degree of freedom for the first 3 ps and a global Nosé-Hoover chain thermostat [50,51,52] for the remaining time. The timestep was set to 0.4 fs. The trajectory was then collected for 15 ps in the NVT ensemble, saving the velocities and coordinates of every step. With the aim of checking the reliability of the AIMD, we calculated the relative energy drift (E${}_{drift}$) of the ‘conserved quantity’ during the production run. This procedure has been proposed in a previous study [53] on some ionic liquids containing highly flexible units like alkyl chains. The slope of the energy drift ($6.01\times 10^{-6}$ a.u./fs) reveals that the dynamics are stable and the chosen parameters for the thermostat and SCF criteria convergence are appropriate.

In order to provide a full characterization of the liquid structure within the range of radial distances accessible to experiments, we have performed a series of simulations with the previously cited (GAFF) using the Gromacs 2019.6 package [54] as the molecular dynamics engine. Partial atomic charges have been obtained using the Restrained Electrostatic Potential (RESP) method [55] by fitting the electrostatic potential for isolated cations and anions at the equilibrium geometry calculated at the HF/6–31G* level. The use of the HF/6–31G* method has been demonstrated to lead to the implicit polarization required in the additive FF model of condensed phase systems, and complies with the rest of the GAFF parameter set, optimized to be compatible with the Cornell et al. families of Amber force fields [56] that implement this charge derivation scheme. Electrostatic interactions were calculated using Particle Mesh Ewald (PME) under periodic boundary conditions and the Linear Constraint Solver (LINCS) algorithm [57] was applied to all bonds involving hydrogen atoms. Cutoff radii for van der Waals and direct-space Ewald interactions were set to 10 Å. Parallelization was carried out with a domain decomposition strategy and the Message Passing Interface (MPI) paradigm.

Topology and coordinate files were prepared using the Amber [58] package (Leap) and all the files were then converted to Gromacs format by using the Babel platform. The initial configurations were generated randomly with the software PACKMOL [59], putting 1000 ion pairs into the cubic box with an initial unit cell edge of 100 Å. Equilibration consisted of $10^{4}$ minimization cycles, by gradually heating the systems at 550 K in the NVT ensemble; the structure was then equilibrated by NPT simulations (2 ns) to remove structural inhomogeneities. Systems were cooled (400 K) and density was calibrated by long NPT simulations (5 ns). Subsequently, the systems were simulated in the NVT ensemble for 3.5 ns with an integration time step of 1 fs and trajectories were collected every 1000 steps.

Structure factors I(q) have been calculated from the molecular dynamics trajectories using the Travis software [60,61]. Theoretical I(q) was then multiplied by *q* and the *q*-dependent sharpening factor M(q) (with nitrogen as the sharpening atom), to obtain a theoretical qI(q)M(q) function comparable to the experimental one. *D(r)* and *Diff(r)* were calculated from theoretical qI(q)M(q) functions by Fourier transform from the reciprocal space (*q*) to the direct one (*r*) following the procedure adopted for the experimental data and described in the supporting information.

## 3. Results and Discussion

### 3.1. QM Ion Pair

H-fumarate and H-maleate anions and the respective choline cation pairs were preliminarily optimized in vacuo to investigate the equilibrium structures of the isolated anions, the coupling geometries and structural changes due to ionic pairing. H-fumarate and H-maleate anions differ in the configuration of the C=C double bond. The equilibrium structure of the trans-isomer, H-fumarate (Figure 1a), is nonplanar with the carboxylate group CO_2_ and nearly orthogonal to the CH=CHCO_2_H moiety. The negative charge is delocalized on the whole carboxylate group with CO bond distances (1.254 and 1.247 Å) and an OCO angle (130.5°) typical of the CO_2_^−^ group. The cis-configuration of the CO_2_^−^ and CO_2_H groups in H-maleate allows the formation of a seven-membered ring for the formation of a strong intramolecular hydrogen bond (Figure 1b). The O⋯HO angle is nearly linear (176.5${}^{\circ }$) and the intramolecular O⋯H distance is very short at 1.336 Å. The H-maleate is 56.5 kJ/mol more stable than H-fumarate and intramolecular hydrogen bonding is largely responsible for this stabilization. Figure 1b shows that the geometry of the H-maleate anion is nearly symmetric although hydrogen is not equally shared between two oxygen atoms; however, it is well localized on one the two oxygen atoms with an OH bond distance of 1.094 Å. The intramolecular proton transfer barrier has been estimated by optimizing the geometry of the isomer $C_{2v}$, where the hydrogen atom occupies a symmetric position. This first order saddle point was found only 10 kJ/mol above the equilibrium structure. This result is largely consistent with previous ab initio[62,63] and experimental [29] studies on H-maleate anions in the gas phase that indicate the free motion of the proton between the oxygen atoms. A rough estimate of the strength of the intramolecular hydrogen bond has been obtained from the study of the conformer of Figure 1c where OH and the carboxylate groups have been oriented to avoid any hydrogen bonding. This open structure was found to be 77.4 kJ/mol above the ring equilibrium structure. Our estimate is comparable with the value (90 kJ/mol) obtained previously for the H-maleate anion in gas phase with the combined study of photoelectron spectroscopy and ab initio calculations [29]. This means that H-maleate in the gas phase assumes quite a rigid structure where intramolecular hydrogen bonding strongly hinders alternative orientations of the OH group.

The structural features of the isolated anions have then been investigated in the presence of the choline cation. The electrostatic potential surface of choline reveals that cations could interact with anions through the OH group, a hydrogen bonding site, as well as through the N(CH_3_)_3_^+^ group, a Coulombic interaction site. Previous studies on choline carboxylate anions [20,21,22] showed that the strong OH⋯O hydrogen bonding between the choline and carboxylate group is a fundamental feature to describe the interactions between the ionic components in the liquid phase. In the case of hydrogen dicarboxylate anions, the choline cation could interact alternatively with the carboxylic group, again leading to hydrogen bonding. In the case of H-fumarate, both the interactions $CO_{2}H\cdots HO$ and $CO_{2}^{-}\cdots HO$ have been considered here and two structures were localized on the potential energy surface for each interaction. Figure 2a,b show the geometries optimized when coupling involves the $CO_{2}^{-}\cdots HO$ groups, whereas Figure 2c,d reproduce the ion pairs formed through $CO_{2}H\cdots HO$ interactions. The binding energies of the ion pair, estimated as the difference between the energy of the complex and the sum of the energies of the cation and anion, are reported in Figure 2. The first result emerging from the calculations is that choline prefers to interact with the carboxylate group than with the carboxylic group; in addition, the coordination is energetically equivalent for the two oxygen atoms of $CO_{2}^{-}$. As a comparison, we report the geometries and binding energies for the most stable structure of the ion pair, as obtained by the MP2/6–311++G** level. The results reproduced in Figure 2 indicate that the structural features of the ion pairs obtained at the B3LYP/6–31++G** level are substantially maintained at the MP2 level. The second important observation is that the binding energies are very large, although the complex formation involves the formation of a single hydrogen bond. This high stabilization could be due to the fact that the binding energy presented here includes all the intermolecular interactions, like hydrogen bonding, Coulombic effects and weaker CH⋯O dipolar interactions. In all the coordination structures of Figure 2, we in fact observe the presence of these CH⋯O contacts. This structural feature is due to the tendency to orient the charged head N(CH_3_)_3_^+^ of the choline cation towards the negative charged carboxylate group of the anion. The cation–anion arrangement in isolated ion pairs therefore allows us to maximize all the intermolecular interactions starting from the highly directional OH⋯O hydrogen bonding to the electrostatic Coulombic interactions between the charged portions N(CH_3_)_3_^+^ and $CO_{2}^{-}$ of the ions ending up to the weaker CH⋯O dipolar interactions. In order to estimate the role of the Coulombic effect on the stability of the complex, additional calculations were carried out on a system where H-fumarate is coupled with a neutral molecule, N,N-dimethylaminoethanol ${\left(CH_{3}\right)}_{2}NCH_{2}CH_{2}OH$. In this complex, we can evaluate the stabilization due to a single hydrogen bond, as for the ion pair, without large contributions derived from electrostatic terms. The stability of such a complex is indeed lower (62 kJ/mol), revealing the dominant role of the Coulombic interactions in the formation of the ion pairs.

As for H-fumarate, as in the case of H-maleate, the ion pairing mainly involves the carboxylate group and the most stable interaction structures are displayed in Figure 3a,b. The OH⋯O distances suggest that hydrogen bonding in the H-maleate ion pair is similar to that calculated in the H-fumarate ion pair when an interaction occurs with O1’, whereas it is weaker when it occurs with O1, the oxygen atom involved in the intramolecular hydrogen bond (for numbering of atoms see Figure 1). Moreover, the interacting geometries suggest that hydrogen bonding in H-fumarate and H-maleate are quite similar, while binding energies calculated for both the structures (Figure 2 and Figure 3) reveal that choline interacts more weakly with H-maleate than H-fumarate and the reason for this is the presence of an intramolecular hydrogen bond in H-maleate. In particular, in the trans-conformation of H-fumarate, the negative charge is mainly localized on the carboxylate group and this gives a large dipole moment to the anion (9.23 D). On the contrary, the π-conjugation through the seven-membered ring allows us to delocalize the negative charge and lower the dipole moment (3.13 D). We expect therefore that the electrostatic component of the binding energy between H-maleate and choline is weaker, as indicated by the values reported in Figure 3. This result is also confirmed at the MP2 level.

As for the isolated anion, the strength of the intramolecular hydrogen bonding of H-maleate has also been investigated in the presence of the choline cation by once again considering the open structure proposed for the isolated anion (Figure 3c). The results obtained suggest that when choline coordinates the carboxylate group, the stability of the intramolecular hydrogen bonding decreases: the open structure was found to be less stable, but the energy difference between the closed and open structure lowers from 77.4 kJ/mol for the isolated anion to 38 kJ/mol for the coordinated anion. This means that the interaction with choline weakens the intramolecular hydrogen bonding, as also revealed by the value of the O⋯H distance, which increases from 1.335 Å to 1.513 Å upon choline coordination.

### 3.2. AIMD Results

The structural features obtained from QM static ionic couples were reconsidered, including the dynamic effects, by AIMD simulations and again with DFT methods. AIMD simulations have been carried out on 10 ion pairs for both the systems. The structure of the anions interacting with choline was initially analyzed through radial distribution functions (RDFs) and dihedral angle distribution functions (DDFs) of some intramolecular geometrical parameters. Conformations of H-fumarate have been considered by the DDFs of the OH, $CO_{2}^{-}$ and $CO_{2}H$ groups, as reproduced in Figure 4a,b. The curves clearly indicate that H-fumarate has a large conformational freedom; the OH group prefers to assume a syn orientation with respect to the C=O bond, whereas the rotation for the carboxylate group is relatively free. On the contrary, the carboxylic group is substantially coplanar with the C=C bond.

As already emerges from the static calculations, H-maleate has, instead, a very rigid conformation, since intramolecular hydrogen bonding deeply affects its dynamic aspects. The DDF of the OH group, reproduced in Figure 4c, shows that the seven-membered ring contains very stable and open structures, similar to those proposed previously, and can be reasonably excluded, at least for the simulation time of our dynamics. Even if hydrogen is mainly covalently bonded to O4, as it just occurs in the ion pair, during dynamics, we can observe the proton transfer from O4 to O1. RDF between O4 and H has, in fact, two peaks—one at values typical of very strong hydrogen bonds (1.4 Å) and a second peak at the covalent bond distance of 1.06 Å (Figure 4d).

In order to investigate the morphology of the systems, at least at short range, hydrogen bonding between cations and anions has been monitored through the RDFs between the hydrogen of the choline OH group and oxygen atoms of the anion. For H-maleate as well as for H-fumarate, the distribution of the intermolecular O⋯H distances indicates that the coordination involves the carboxylate group (O1 and O1’) and the carboxylic group through O4’. The contribution for both the oxygen atoms O1 and O1’ is comparable for H-fumarate (Figure 4e) as a consequence of the fact that the carboxylate group has a nearly symmetric charge distribution. The O⋯H distance is in line with those determined for the static ion pair. Some differences are found in the case of H-maleate: the choline cation shows a higher probability to bind O1’ than O1 (Figure 4f) since intramolecular hydrogen bonding gives asymmetry to the carboxylate group. In addition, the peaks of the RDFs of H-maleate are slightly larger than those of H-fumarate, in agreement with the fact that hydrogen bonding between cations and anions is less directional when intramolecular hydrogen bonding is formed. Figure 4g also shows the RDFs between the nitrogen of choline and the oxygen of the anions. Such RDFs show large peaks at about 4 Å. This correlation, quite common in choline carboxylate ionic liquids [20,21,22], is due to the electrostatic interaction between the negatively charged oxygen atoms of the anions and the N(CH_3_)_3_^+^ portion of the cations. These additional interactions act as a docking force, determining the local morphology of the ion–ion interaction. On the other hand, the tendency to favourably orient cations and anions to maximize the reciprocal interactions was already found from the structures of the small static ion pairs discussed above. Bearing in mind that our AIMD simulations are too short to give a full description of dynamics of hydrogen bonding, the results obtained clearly show that H-maleate forms intramolecular hydrogen bonds and keeps its closed structure during the time of the dynamics. This structural feature can have an important role in the distribution of the ionic species in the liquid. As normally occurs in ionic liquids, the first neighbor contacts are those between cations and anions and the narrow distribution of O⋯H values at about 1.8 Å provides clear evidence that cations and anions form strong hydrogen bonds through carboxylate and carboxylic groups. One interesting aspect emerges when we analyse the anion–anion distribution distances in the two systems. In the trans-conformation, H-fumarate can establish hydrogen bonds between $CO_{2}^{-}$ and COOH groups, as shown by the RDF O⋯O distances reproduced in Figure 4h. The structure of [Ch][H-fumarate] liquid should be therefore characterized by a complex pattern: part of the anions are strongly coupled with cations by electrostatic and hydrogen bonding; on the other hand, the presence of donor acceptor sites on the anions makes them able to interact with each other by hydrogen bonds. No correlation between anions was found for [Ch][H-maleate] since intramolecular hydrogen bonding hinders any interaction between anions. Similarly, no correlation was found between choline cations.

### 3.3. X-ray Results and MD Simulations

Since our AIMD models are clearly too small to provide long range structural features and the application of ab initio molecular dynamics on systems larger than those proposed here is still prohibitive, the [Ch][H-maleate] and [Ch][H-fumarate] liquids were investigated through classical molecular dynamics. Despite classical force fields not always accurately describing protic ionic liquids [21,64,65], we decided to use the GAFF force field anyway since it covers almost all the organic species without the need for further parametrization and it gave good results for some systems like choline salicylate liquid [22]. As discussed for the AIMD results, the structure of our liquids was analyzed by monitoring some important intermolecular distances, starting from the hydrogen bond ones. The presence of strong hydrogen bonding between the choline and oxygen atoms of the anions is confirmed by the classical simulations. RDFs of the H⋯O contacts reproduced in Figure 5a,b indicate that the interaction mainly involves the carboxylate group and, to a lesser extent, also the O4’ atom of the carboxylic group. A remarkable difference between ab initio and classical models is the fact that the O1 and O1’ atoms of the carboxylate group are equally bonded to choline both in H-fumarate (Figure 5a) and H-maleate (Figure 5b). Thus, the small asymmetry in the hydrogen bonding acceptor properties of the carboxylate oxygen of H-maleate emerging from ab initio results is not appreciable from the classical force field. The equivalence of the two hydrogen bonds is consistent with the fact that the force field here applied describes the carboxylate group with nearly symmetric interaction potentials. The structure of the liquids presents an alternating pattern of anions and cations, as witnessed by the structured peak in the N⋯O intermolecular RDF at about 4 Å (Figure 5c,d). For both the systems, the $CO_{2}^{-}$ group coordinates the cationic charged head $N{\left(CH_{3}\right)}_{3}^{+}$ better than COOH for stronger electrostatic effects. It is interesting to note that the coordination of the $N{\left(CH_{3}\right)}_{3}^{+}$ portion to the anion leads to the formation of CH⋯O contacts with all the oxygen atoms and Figure 5e shows the RDFs of one of the hydrogen atoms of the methyl groups with all the oxygen atoms of the H-maleate anion.

It is worth noting that the anion–anion distribution is different for the two liquids, as already found from our AIMD simulations. Figure 5f confirms that, in [Ch][H-fumarate] liquid, in addition to the alternation of cations and anions, anion–anion interactions driven by hydrogen bonding between $CO_{2}^{-}$ and COOH groups are expected. Such structural feature can instead be excluded in the [Ch][H-maleate] system.

The structure of the two liquids has been studied experimentally through X-ray diffraction and, in Figure 6, we report the X-ray structure function of each system (upper panel) and the complementary *Diff(r)* function (lower panel). The interpretation of the diffraction pattern is not easy since the experimental data are the averaged results of all the inter-atomic and intra-atomic distances of the systems. In addition, the comparison between the two systems shows that the curves are indeed very similar. As known, the pattern above 5 Å${}^{-1}$ corresponds roughly to intramolecular distances, whereas the intermolecular distances are expected to give rise the peaks below this value. In this region, the principal peak, at about 1.3 Å${}^{-1}$, which correlates with distances between 3 and 6 Å, is due to the alternating pattern of cations and anions in the liquid. Figure 6 shows that the main peak of [Ch][H-maleate] is slightly larger and has the maximum at lower *q* with respect to the [Ch][H-fumarate]. This feature could be due the fact that the closed structure of H-maleate reduces its ability to coordinate cations by hydrogen bonds, giving a less compact liquid structure than H-fumarate. Consistently, the peak at about 6 Å${}^{-1}$ occurs at higher *q* for H-maleate since its closed geometry gives a peak centered at smaller intramolecular distances. These slight differences and the structural features emerging from our MD simulations suggest that the open structure of [Ch][H-fumarate] could give a less structured liquid.

The differences are more marked in the shoulder appearing at the right of the main peak: [Ch][H-maleate] shows a peak at about 3.5 Å${}^{-1}$, whereas [Ch][H-fumarate] shows one at 2.8 Å${}^{-1}$. The first peak correlates with distances at 1.8 Å and the second with contacts at 2.2 Å in the direct space. Such peaks can be due to both inter- and intramolecular distances. Such features are typical of choline carboxylate ionic liquids and have been previously considered as fingerprints of hydrogen bonding interactions [26]. Similarly, the *Diff(r)* curves of the two systems show a different pattern, particularly in the region between 4 and 6 Å: [Ch][H-maleate] shows two distinct peaks at 4.4 Å and 5.6 Å whereas [Ch][H-fumarate] shows a wide and single peak around 5 Å. One of the reasons for this different pattern could be found in the different conformations of the H-maleate and H-fumarate anions. The presence of two peaks in the [Ch][H-maleate] could be consistent with the distribution of the O⋯O intramolecular distances in the cis-isomer; the anti-conformation instead would give rise to intramolecular distances very similar to each other, as summarized in Figure 6. In addition, strong peaks between 4 and 6 Å might be attributed to cation–anion as well as anion–anion correlations, as indicated by RDFs expected in this region (Figure 4 and Figure 5). Another explanation of these differences could be tentatively found by considering that the anion–anion distribution predicted by MD simulations for [Ch][H-fumarate] could produce peaks broader than those observed for [Ch][H-maleate], where the only correlation derives from the alternating cationic–anionic pattern. A validation of the theoretical results here presented can be obtained by comparing the findings from the MD simulation with the measured X-ray diffraction pattern. The results are presented in Figure 7. The agreement between the qI(q)M(q) experimental curves and the same functions calculated from MD simulations is quite good across all *q* values, with a few minor differences in the region between 3 and 4 Å${}^{-1}$. In particular, MD simulations reproduce the shoulders of the main peak well, but introduce a secondary peak at about 4 Å${}^{-1}$ for [Ch][H-fumarate] and at 4.8 Å${}^{-1}$ for [Ch][H-maleate], which is not present in the experimental curves and that correlate with some intramolecular distances that were not fully reproduced by GAFF force field. It is interesting to observe that the complementary *Diff(r)* functions are satisfactory reproduced in both the intra- and intermolecular structural range with some differences in the region of the hydrogen bonding, probably due to the limit of the GAFF force field in reproducing this critical spectral region [21,64]. Another discrepancy is observed in the region of the long distances for [Ch][H-fumarate] where the theoretical curve is quite smooth. This means that our model describes [Ch][H-fumarate] as a liquid that is less structured than [Ch][H-maleate] since the alternation between cations and anions may be perturbed by the presence of anion–anion interactions that give a more random distribution of the ionic components. It is likely that our force field overestimates this effect and, consequently, the agreement in this region worsens.

### 3.4. IR and Raman Spectra

The vibrational properties of the two liquids were studied through IR and Raman spectroscopy, whose patterns are shown in Figure 8 and Figure 9. The assignment of the main bands was made on the basis of DFT calculations as well as by considering vibrational properties of choline carboxylate ionic liquids previously measured [20,21,22]. IR and Raman spectra of the H-fumarate and H-maleate anions, choline cation and the respective ion pairs considered in the most stable structures were calculated at the B3LYP/6–31++G** level. Vibrational modes were analyzed on the basis of the calculated PED and the assignments are reported in Appendix A. Although the theoretical spectra here presented do not actually describe the liquid phase but only ion pairs, the main effect on the vibrational modes due to hydrogen bonds can be estimated from our models anyway. The analysis of the normal modes indicate in fact that the vibrations are mainly localized on each ionic component with coupling indeed being slight and limited to the very low frequency modes. Therefore, it is quite easy to separate the vibrations of the choline cation from those of the anions. In addition, the theoretical spectra show that the choline bands are found to be substantially unchanged in the two ion pairs, suggesting that the coupling effect on the vibration modes of choline is indeed similar in both systems. The observed and calculated frequencies are summarized in Table 1 and Table 2. The high-frequency zone of the IR spectra is largely dominated by the broad OH stretching absorption, with weaker bands originated by the CH stretching modes. On the contrary, the CH stretching vibrations are clearly evident in the Raman spectra and give rise to three intense bands, at about 3036, 2980 and 2930 cm${}^{-1}$, assigned to vibrations of the $CH_{3}$ and $CH_{2}$ groups of choline and CH bonds of the anions. The theoretical spectra show that this zone is very similar for the two systems; the only difference is expected for the OH stretching mode. Its frequency is strongly affected by intramolecular hydrogen bonding and by its strength. For example, in the isolated anion, where the proton is nearly symmetric between the oxygen atoms, this mode is strongly coupled with the CO stretching motions and its frequency is very low (1814 cm${}^{-1}$). In the ion pair, the strength of the intramolecular interaction lowers and the relative frequency increases to 2805 cm${}^{-1}$. This vibration has been measured at 2833 cm${}^{-1}$ in the choline salicylate liquid, where the OH group of the anion is involved in a strong intramolecular hydrogen bond. However, we are aware that this vibration could be strongly affected by all the interactions that the anions may make in the liquid phase and its frequency is not accurately reproduced by a single ion pair. The IR spectrum of H-maleate shows broad and very weak absorptions in this region that we can hardly assign to this mode.

At a lower frequency, we expect the bands of the stretching modes of the COOH and $CO_{2}^{-}$ groups and the C=C bond. The C=C stretching vibration gives rise to intense absorptions in the Raman spectra of [Ch][H-fumarate] (1661 cm${}^{-1}$) and [Ch][H-maleate] (1623 cm${}^{-1}$), in agreement with the calculated values at 1698 and 1669 cm${}^{-1}$, respectively. The lower vibrational frequency for the C=C bond of H-maleate is expected as a consequence of the intramolecular hydrogen bond: the formation of the seven-membered ring allows for π-conjugation through the C=C bond and the lowering of the relative stretching frequency. Such bands are less intense in the IR spectra and are measured at 1640 and at 1610 cm${}^{-1}$, respectively. The C=O stretching mode of the COOH group is observed at 1706 (IR) and 1700 (Raman) for H-fumarate and at 1712 (IR) and 1694 (Raman) for H-maleate. The C–O stretching mode of the COOH group can be assigned to the band at 1189 cm${}^{-1}$ for [Ch][H-fumarate], whereas it is not easy to find the relative absorption in the [Ch][H-maleate] spectra.

The stretching vibrations of the $CO_{2}^{-}$ group can be classified as CO asymmetric and CO symmetric stretching and the relative bands are observed in the IR spectrum at 1582 and 1389 cm${}^{-1}$ for [Ch][H-fumarate] and 1580 and 1361 cm${}^{-1}$ for [Ch][H-maleate], respectively. The CO symmetric vibrations are measured in the Raman spectrum at 1384 cm${}^{-1}$ for [Ch][H-fumarate] and at 1382 cm${}^{-1}$ for [Ch][H-maleate]; the asymmetric vibration is weaker and absorption is appreciable only for the [Ch][H-fumarate] at 1580 cm${}^{-1}$. It is interesting to note that these two absorptions are measured at frequencies that are very similar in the two anions, as expected from the DFT values calculated for the ion pairs. As known [20], hydrogen bonding can be analyzed also by looking at the stretching frequencies of the $CO_{2}^{-}$ group. An isolated carboxylate group has a symmetric geometry and the extent of coupling between symmetric and asymmetric CO stretching is high. Hydrogen bonding induces asymmetry in the $CO_{2}^{-}$ structure, lowering the coupling between the two modes. This effect is well summarized by the difference between the asymmetric and symmetric CO stretching frequencies, ΔνCO. In the H-fumarate-isolated anion where the carboxylate is free, ΔνCO is 297 cm${}^{-1}$, whereas, in the ion pair, where carboxylate forms hydrogen bonding, the frequencies of the two modes approach each other and ΔνCO lowers to 228 cm${}^{-1}$. A similar value, 239 cm${}^{-1}$, is calculated for the choline H-maleate ion pair where carboxylate is involved in intramolecular as well as intermolecular hydrogen bonds. On the basis of our assignments, the ΔνCO obtained from the experimental frequencies for [Ch][H-fumarate] (200 cm${}^{-1}$) and [Ch][H-maleate] (200 cm${}^{-1}$) is very similar for the two liquids, suggesting that the carboxylate group is perturbed similarly in both systems.

Some vibrational modes are more sensitive to hydrogen bonding—in particular the in-plane bending, δOH, and out-of-plane deformation τOH—of the OH group. The δOH motion is appreciably coupled with the in-plane bending vibrations of the CH groups; however, the PED suggests that the band at 1292 cm${}^{-1}$ in the IR spectrum of [Ch][H-fumarate] could be assigned to a mode where the δOH component is prevalent. Intramolecular hydrogen bonding strongly hinders this motion in the choline H-maleate and its frequency is measured at much higher values (1588 cm${}^{-1}$). For the same reason, the OH torsion, τOH, increases its frequency from 630 cm${}^{-1}$ in H-fumarate to 1001 cm${}^{-1}$ in H-maleate. Further in-plane and out-of-plane vibrations of the anions are presented in Table 1 and Table 2.

Choline bands were measured in some carboxylate choline liquids [20,21,22] or in systems where choline is weakly coupled with anions [67] and we find them substantially unchanged in our liquids. They are reported in Table 1 and Table 2.

## 4. Conclusions

The structure of two ionic liquids where the choline cation is coupled with two isomers of the monoanion of the maleic acid ($HOOCCH=CHCOO^{-}$) has been studied by energy-dispersive X-ray diffraction, IR and Raman spectroscopy and theoretical models. In the cis-conformation (H-maleate), the anion forms an intramolecular hydrogen bond that hinders the internal rotations of the OH group; this group is instead free to rotate in the trans-conformation (H-fumarate). Different theoretical models have been proposed to describe the liquids and the results have been compared with the experiments. Single ion pairs have been studied by DFT methods and small portions of the liquids have been modeled by AIMD methods and the results indicate that choline and anions are connected by hydrogen bonds. All calculations indicate that the choline cation prefers to connect mainly to the carboxylate group through OH⋯O interactions in both the compounds. Intermolecular bond distances are quite similar in the two systems, although the interaction energy is higher for H-fumarate. In this case, the negative charge is strictly localized on the $CO_{2}^{-}$ group and the electrostatic component of the interaction energy is therefore higher. MD simulations obtained with the GAFF force field confirm the cation anion distribution found with AIMD methods: the proton of choline is sharply localized around the oxygen atoms of the carboxylate groups and the N(CH_3_)_3_^+^ charged head of choline is oriented towards the negative portion of the anion. An intramolecular hydrogen bond of H-maleate is found in the gas phase as well as in the liquid phase and its force is weakly affected by choline coordination. The trans-conformation of H-fumarate allows further interactions between COOH and $CO_{2}^{-}$ of the anion and gives rise to correlations between anions in the radial distribution functions. No anion–anion correlation has been found for H-maleate since intramolecular hydrogen bonding inhibits additional interactions between anions. The X-ray experimental pattern has been reproduced by MD simulations and structure functions and the complementary *Diff(r)* curves obtained by calculations are in good agreement with the X-ray experiments. IR and Raman spectra have been measured for both systems and some bands have been assigned on the basis of DFT frequency calculations. The vibrational stretching modes of the $CO_{2}^{-}$ group are quite similar in the two systems, revealing that the carboxylate group is perturbed similarly in both the anions. This is consistent with the structural features obtained from QM results that indicate very similar coupling geometries for both systems. The intramolecular hydrogen bond in H-maleate is confirmed by the frequency of some vibrational motions, like OH group torsion or OH bending.

## Figures and Tables

**Figure 1 molecules-25-04990-f001:**
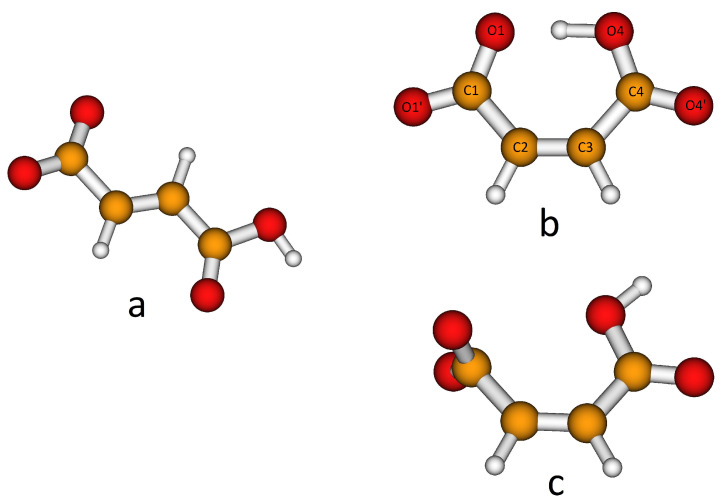
Equilibrium structures of the H-fumarate (**a**) and H-maleate (**b**) anions and open structure of the H-maleate anion (**c**).

**Figure 2 molecules-25-04990-f002:**
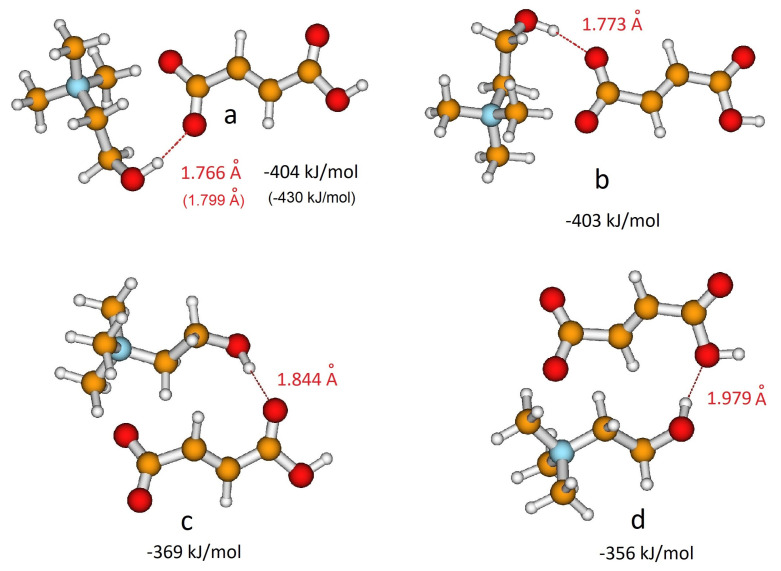
Different coordination structures, (**a**), (**b**), (**c**) and (**d**), of H-fumarate anion and choline cation. Hydrogen bonding distance and binding energy are evaluated from B3LYP/6–311++G** calculations. Values in parentheses are obtained from MP2/6–311++G** calculations.

**Figure 3 molecules-25-04990-f003:**
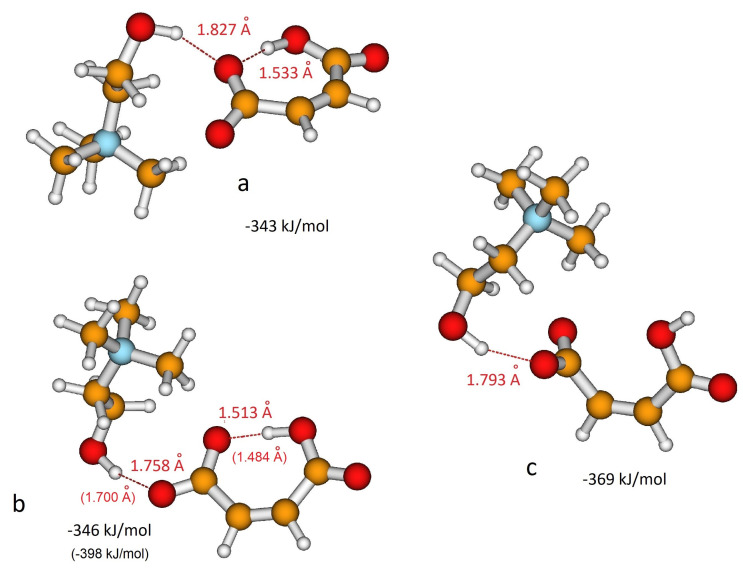
Different coordination structures, (**a**), (**b**) and (**c**), of H-maleate anion and choline cation. Hydrogen bonding distance and binding energy are evaluated from B3LYP/6–311++G** calculations. Values in parentheses are obtained from MP2/6–311++G** calculations.

**Figure 4 molecules-25-04990-f004:**
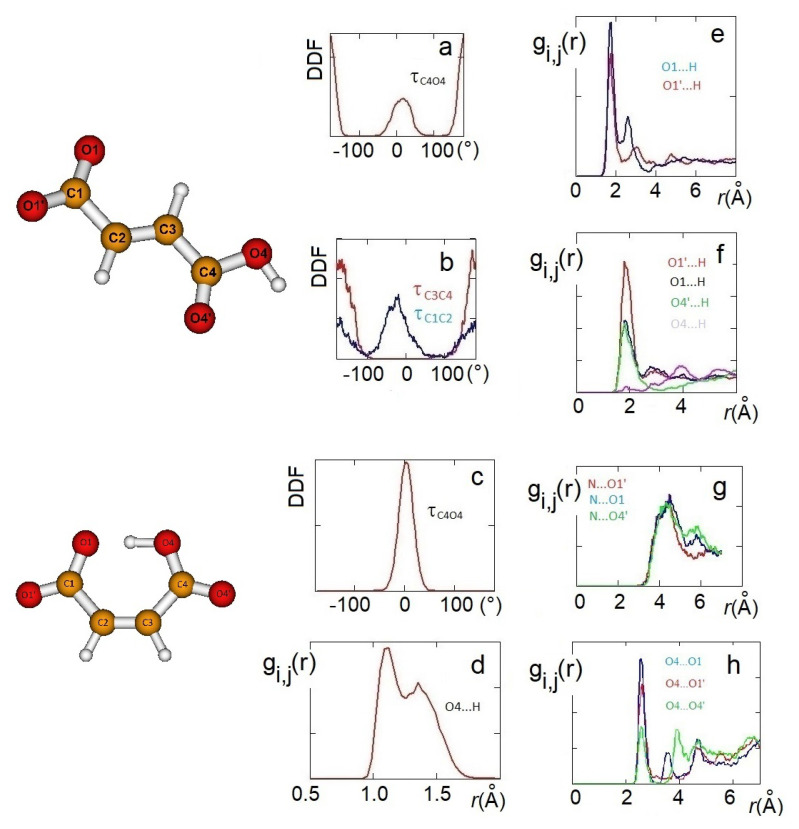
AIMD results: HO4C4C3 dihedral angle distribution functions (DDFs) of [Ch][H-fumarate] (**a**) and [Ch][H-maleate] (**c**); O4C4C3C2 and O1C1C2C3 DDFs of [Ch][H-fumarate] (**b**). Radial distribution function (RDF) of the intramolecular O⋯H distance in [Ch][H-maleate] (**d**); RDFs of the H⋯O cation–anion distances in [Ch][H-fumarate] (**e**) and [Ch][H-maleate] (**f**); RDFs of the N⋯O cation–anion distances in [Ch][H-maleate] (**g**); RDFs of the O⋯O anion–anion distances in [Ch][H-fumarate] (**h**).

**Figure 5 molecules-25-04990-f005:**
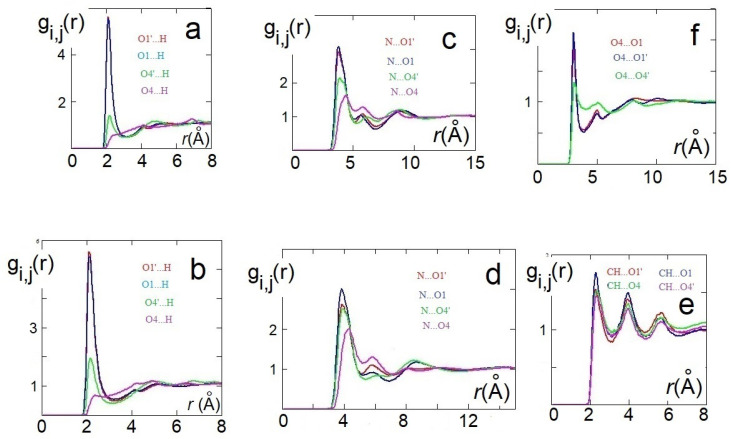
RDFs of the H⋯O, N⋯O and CH⋯O cation anion distances in [Ch][H-fumarate] (**a**,**c**,**e**) and [Ch][H-maleate] (**b** and **d**); RDFs of the O⋯O anion–anion distance in [Ch][H-fumarate] (**f**) from classical molecular dynamics (MD) simulations.

**Figure 6 molecules-25-04990-f006:**
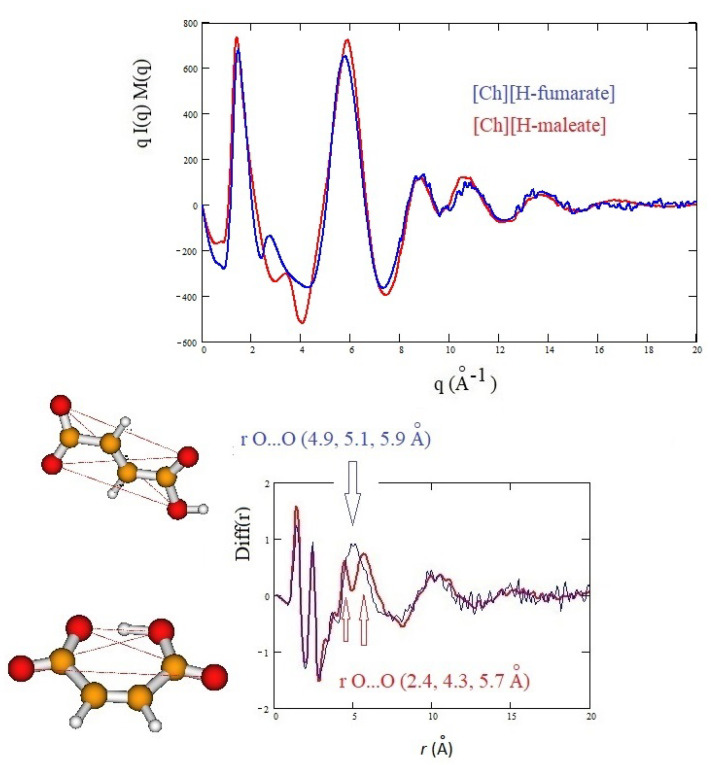
Experimental *qI(q)M(q)* and *Diff(r)* curves of [Ch][H-maleate] and [Ch][H-fumarate]; figure shows the intramolecular O⋯O distances calculated from density functional theory (DFT) geometries of the two anions.

**Figure 7 molecules-25-04990-f007:**
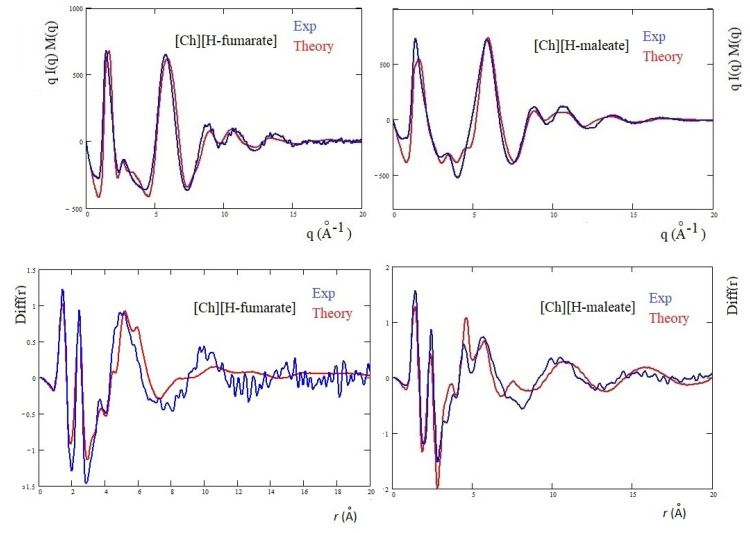
Theoretical *qI(q)M(q)* and *Diff(r)* of H-maleate and H-fumarate in comparison with experiment.

**Figure 8 molecules-25-04990-f008:**
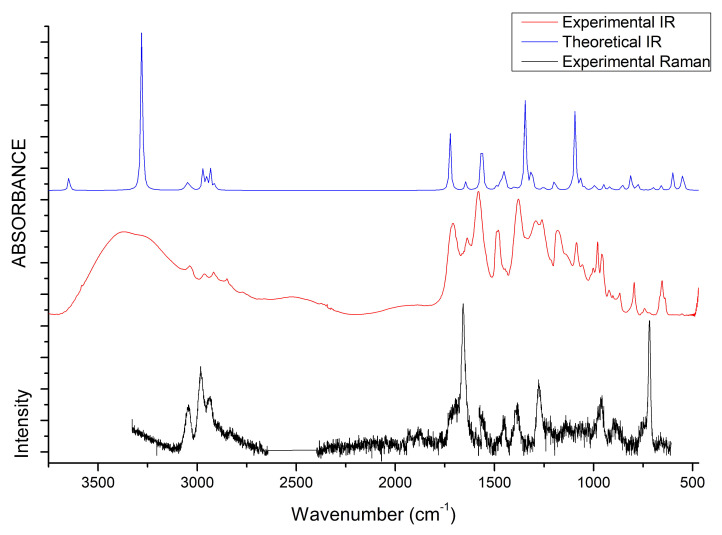
IR and Raman spectra of [Ch][H-fumarate] liquid. B3LYP values are scaled for 0.96 [66].

**Figure 9 molecules-25-04990-f009:**
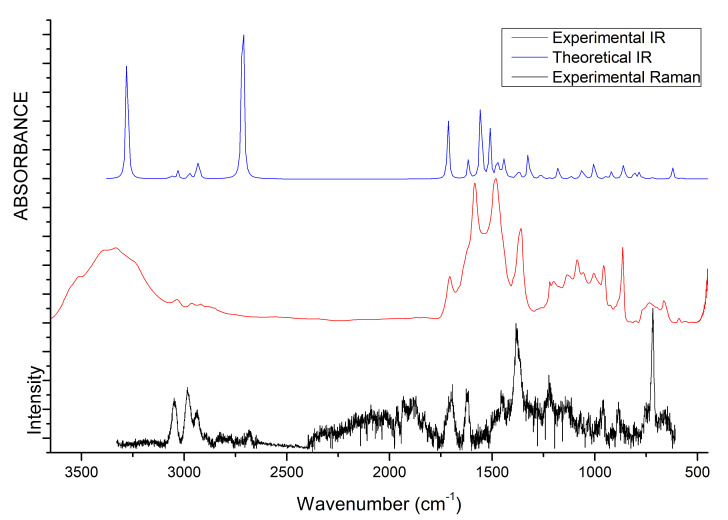
IR and Raman spectra of [Ch][H-maleate] liquid. B3LYP values are scaled for 0.96 [66].

**Table 1 molecules-25-04990-t001:** Observed (IR and Raman) and calculated (B3LYP/6–311++G**) vibrational frequencies [cm${}^{-1}$] [Ch][H-fumarate].

Experiment		Theory [a]			Assignment [b,c]
**IR**	**Raman**	**Frequency**	**Intensity**	**Intensity**	
			**IR (km/mol)**	**Raman (Å${}^{4}$/amu)**	
630		572	78	1	τCO
651		620	97	3	δOCC, δOCO
750	716	723	21	9	ωOCO, δOCO
796		803	30	0	ωOCO
850	850	837	102	3	τCO choline
920		884	42	8	νCN choline
954	960	980	31	5	νCN choline
980		1026	39	0	γCH
1085		1101	55	3	νCO choline
1189		1129	441	0	νCO
1251		1237	64	7	δHCC
	1277	1288	9	35	δHCC
1292		1355	122	4	δHCC, δHOC
1389	1384	1388	335	49	ν $CO_{2}^{-}$ ${}_{sym}$
	1453	1492	59	33	δHCH choline
1485		1505, 1498	44, 48	3, 11	δHCH choline
1582	1580	1615	407	10	ν $CO_{2}^{-}$ ${}_{asym}$
1637	1661	1698	50	332	νC=C
1706	≈1700	1780	317	52	νC=O
2930	2938	3048, 3030	36, 115	115, 169	νCH${}_{3}$ choline
2980	2983	3069	114	350	νCH${}_{2}$ choline
		3164, 3152, 3148	18, 25, 15	62, 71, 83	νCH${}_{3}$ choline
3036	3050 {				
		3200, 3180	0, 2	32, 55	νCH

${}^{\left[a\right]}$ Frequencies calculated for the ion pair. ${}^{\left[b\right]}$ Frequency values shown in red represent bands assigned to the choline cation. ${}^{\left[c\right]}$ν, stretching; δ, bending; τ, torsion; ω, wagging; ρ, rocking; γ, out-of-plane deformation.

**Table 2 molecules-25-04990-t002:** Observed (IR and Raman) and calculated (B3LYP/6–311++G**) vibrational frequencies [cm${}^{-1}$] [Ch][H-maleate].

Experiment		Theory [a]			Assignment [b,c]
**IR**	**Raman**	**Frequency**	**Intensity**	**Intensity**	
			**IR (km/mol)**	**Raman** (**Å${}^{4}$/amu)**	
594		595	6	1	ωOCO, τCC
664		641	66	2	νCC, δOCC, δOCO
734	716	744	10	14	νCN choline
865	850	834	51	0	τCO choline
910		893	46	7	νCN choline
961	960	950	24	1	νCN choline
1001		1039	70	2	τCO
1050		1035	24	2	γCH
1085		1100	49	4	νCO choline
1127	1136	1154	13	3	ρCH3 choline
1200		1216	29	6	ρCH3 choline
1209	1222	1218	51	8	δHCC
1361	1382	1368	153	49	ν $CO_{2}^{-}$ ${}_{sym}$
1482	1460	1489, 1517	85, 49	10, 1	δHOC, δHCC choline
1588		1561	293	2	δHOC
≈1580		1607	563	2	ν $CO_{2}^{-}$ ${}_{asym}$
	1623	1669	112	126	νC=C
1712	1694	1772	353	121	νC=O
	2680-2650	2804	1495	46	νOH
2930	2938	3033	117	226	νCH${}_{3}$ choline
2980	2983	3070, 3076	8, 20	132, 170	νCH${}_{2}$ choline
		3160, 3156	6, 11	52, 100	νCH${}_{3}$ choline
3036	3050 {				
		3175, 3154	12, 0	155, 50	νCH

${}^{\left[a\right]}$ Frequencies calculated for the ion pair. ${}^{\left[b\right]}$ Frequency values shown in red represent bands assigned to the choline cation. ${}^{\left[c\right]}$ν, stretching; δ, bending; τ, torsion; ρ, rocking; γ, out-of-plane deformation.

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
