# Peer review of "Choline Hydrogen Dicarboxylate Ionic Liquids by X-ray Scattering, Vibrational Spectroscopy and Molecular Dynamics: H-Fumarate and H-Maleate and Their Conformations"

_molecules, 2020, doi:10.3390/molecules25214990_

Round 1

Reviewer 1 Report

I send them in the enclosed attachment

Author Response

.

Reviewer 2 Report

The authors have used the MD and QM techniques to study the structure and spectrum of two ionic liquids. The role of hydrogen bonding on the overall stabilization is explored. The manuscript may be published after a few minor corrections :

  1. The most significant clarifications pertaining to the QM calculations is that the authors must justify the choice of functional employed for the given system. Moreover, what is the trend based on other electronic structure methods? Similarly, how do their results improvize beyond already reported work.

2.  Secondly, the stabilization energy for the systems in Figure 2 and 3 is very high although the complex formation is virtually due a single hydrogen bond. This is not clear and as authors try to justify in line 164 it looks more of a hypothesis.

3. In Fig 7, the authors find good agreement with all cases except for [Ch][H-fumarate]. Especially the long order trends in theory are devoid of structural information and a straight curve.  Is this well-converged result and the discrepancy analyzed?

4. Figure 4 and 5 should be redrawn with the x-axis and plots being more clear and readable. Its not possible to understand them in the current form without pushing your eyes to a limit.

5. For the IR spectra calculations, its not clear if the calculations are done with QM for a  molecule or based on the trajectories obtained from AIMD/classical MD.  How have the authors got the IR spectra for liquid IL using QM ? If it is using AIMD what is peak

characterization algorithm?

6. A cutoff 320 Ry is not suitable for the standard AIMD simulations especially when the system is highly flexible/charged and involved conformational features. The authors should justify the computational set up. Maybe the energy convergence for the given setup may be provided which should be already available in the existing simulation log files. This will help the reader gauge the accuracy of simulations.

Minor comments: The method sections for MD must be rewritten.

Line 91 : initial configuration and not snapshot

Line 116: Authors should be more specific instead of using statements like "several NVT simulations". This helps no way to redo the reported simulations.

Line 117: timestep is always 0.1fs: Is it changed and any time during simulations. More efficient writing efforts needed.

Line 85 : Citation for CP2K should be updated: https://doi.org/10.1063/5.0007045

Line 41 : "can really affect the microscopic structure"  instead of "whole structure".

I will also like the authors to expand the literature survey beyond IL for their computational works and include studies on condensed phase like

https://doi.org/10.1021/acs.chemrev.5b00640

https://doi.org/10.1038/s41598-018-35357-9

10.1039/C8CP05880F

Author Response

.
